# Ten-year natural history of visual function in Japanese patients with Leber hereditary optic neuropathy: A retrospective cohort study

Yasuyuki Takai[1]*, Akiko Yamagami[1], Mayumi Iwasa[1,2], Kenji Inoue[1], Ryoma Yasumoto[3], Hitoshi Ishikawa[4], Masato Wakakura[1]

**1** Department of Ophthalmology, Inouye Eye Hospital, Tokyo, Japan, **2** Department of Ophthalmology, Michikusa Eye Clinic Nakamurabashi, Tokyo, Japan, **3** Department of Clinical Laboratory, Kitazato University Hospital, Kanagawa, Japan, **4** Department of Orthoptics and Visual Science, School of Allied Health Sciences, Kitazato University, Kanagawa, Japan

* tkysyk3@gmail.com

## Abstract

Leber hereditary optic neuropathy (LHON) is a mitochondrial optic neuropathy that typically causes severe bilateral central vision loss. Although several natural-history studies have reported visual outcomes, long-term trajectories beyond 5 years remain incompletely quantified, particularly in Asian cohorts. We performed a single-center retrospective cohort study at a tertiary eye hospital in Japan. Among 174 genetically confirmed LHON patients, we identified 27 patients (53 eyes) who presented within 6 months of onset and were followed continuously for ≥10 years. Best-corrected visual acuity (BCVA, logMAR) was collected from onset to 120 months. Piecewise linear mixed-effects models with patient- and eye-level random effects were fitted over three phases (Acute, < 12 months; Chronic, 12–60 months; Late Chronic, 60–120 months). Kaplan–Meier analyses estimated time to achieving BCVA ≤1.6 and ≤1.3 logMAR. BCVA worsened rapidly after onset, followed by gradual improvement and then an approximately horizontal course. During the Chronic phase, BCVA showed a small but statistically significant improvement (overall −0.030 logMAR/year, 95% confidence interval (CI) −0.045 to −0.015; −0.025 logMAR/year, 95% CI −0.044 to −0.007 in a pre-specified m.11778G > A subgroup). In the Late Chronic phase, estimated slopes were close to zero (+0.003 logMAR/year, 95% CI −0.007 to +0.013 overall), with wide CIs compatible with small long-term improvement or deterioration. By 10 years, the cumulative probabilities of achieving BCVA ≤1.6 and ≤1.3 logMAR were approximately 45% and 26%, respectively. In this Japanese LHON cohort with ≥10 years of continuous follow-up, BCVA showed limited improvement during the early chronic period, and we did not detect a clear directional trend thereafter. Given the modest sample size and selection of long-term attendees, these estimates should be interpreted as descriptive, phase-specific benchmarks rather than definitive evidence of long-term stability or ethnic differences.

**Data availability statement:** All relevant data are within the manuscript and its Supporting information files (S1–S3 Datasets).

**Funding:** The author(s) received no specific funding for this work.

**Competing interests:** The authors have declared that no competing interests exist.

## Introduction

Leber hereditary optic neuropathy (LHON) is caused by a mitochondrial DNA disorder [1], with m.11778G>A, m.14484T>C, and m.3460G>A mutations accounting for 90% of cases. Although predominantly affecting young men, it can also occur in women, older adults, and children. The visual prognosis is generally poor, with visual acuity often declining to >1.0 logMAR. Regarding the long-term course of visual function, Lam et al. reported changes in visual acuity in 44 patients with LHON [2]. The mean time from vision loss onset to study enrollment was 97.7 months (range: 0.8 months to 45 years). By monitoring the visual acuity every 6 months for 36 months, they characterized the visual course of LHON. They noted that visual deterioration typically occurs within the first year of onset, spontaneous recovery occurs in approximately 18% of cases, and visual recovery can occur as late as 72 months after onset. However, the participants were not uniformly enrolled within the first year of onset; the time since onset varied widely at enrollment.

Recently, disease-modifying therapies for LHON, including oral idebenone [3] and intravitreal gene therapy [4], have been investigated in controlled clinical trials. Although key studies have used randomized placebo- or sham-controlled or comparative designs, the rarity of LHON and heterogeneity in onset and follow-up make it difficult in practice to assemble control groups at adequate scale. Data on long-term visual trajectories in Asian LHON cohorts remain limited, and cross-cohort differences may reflect variation in mitochondrial genetic background (e.g., haplogroup distributions), environmental exposures, and healthcare access.

Because the continuous long-term course from onset—particularly beyond the chronic phase—remains insufficiently characterized, we aimed to (i) describe the 10-year natural history of BCVA in a Japanese LHON cohort; (ii) estimate the chronic-phase slope using piecewise LMMs aligned with prior natural-history definitions; and (iii) assess consistency in a predefined subgroup defined as onset age≥15 years and m.11778G>A subgroup [5]. In contrast to prior natural-history reports that often include heterogeneous enrollment timing and variable follow-up completeness, our study focuses on a near-onset cohort (presentation within 6 months) with continuous follow-up for ≥10 years at a single tertiary center, enabling phase-specific longitudinal estimates over the chronic and late chronic periods.

## Materials and methods

We conducted a single-center retrospective cohort study at Inouye Eye Hospital, including all genetically confirmed LHON patients seen between April 1990 and March 2020 (n=174). Eligible patients carried a primary LHON mutation (m.11778G>A, m.14484T>C, or m.3460G>A), presented within 6 months of visual loss onset, had no ocular comorbidity likely to affect visual function (e.g., cataract or glaucoma), and were followed continuously for ≥10 years with at least annual BCVA measurements. After applying these patient-level criteria, 27 patients were included. At the eye level, among 54 eyes from these 27 patients, one eye developed retinal vein occlusion during follow-up and was excluded from BCVA analyses, resulting in 53 eyes for the primary analyses (Fig 1). No patients received idebenone or gene therapy during follow-up. Low-dose, non-specific off-label supportive medications

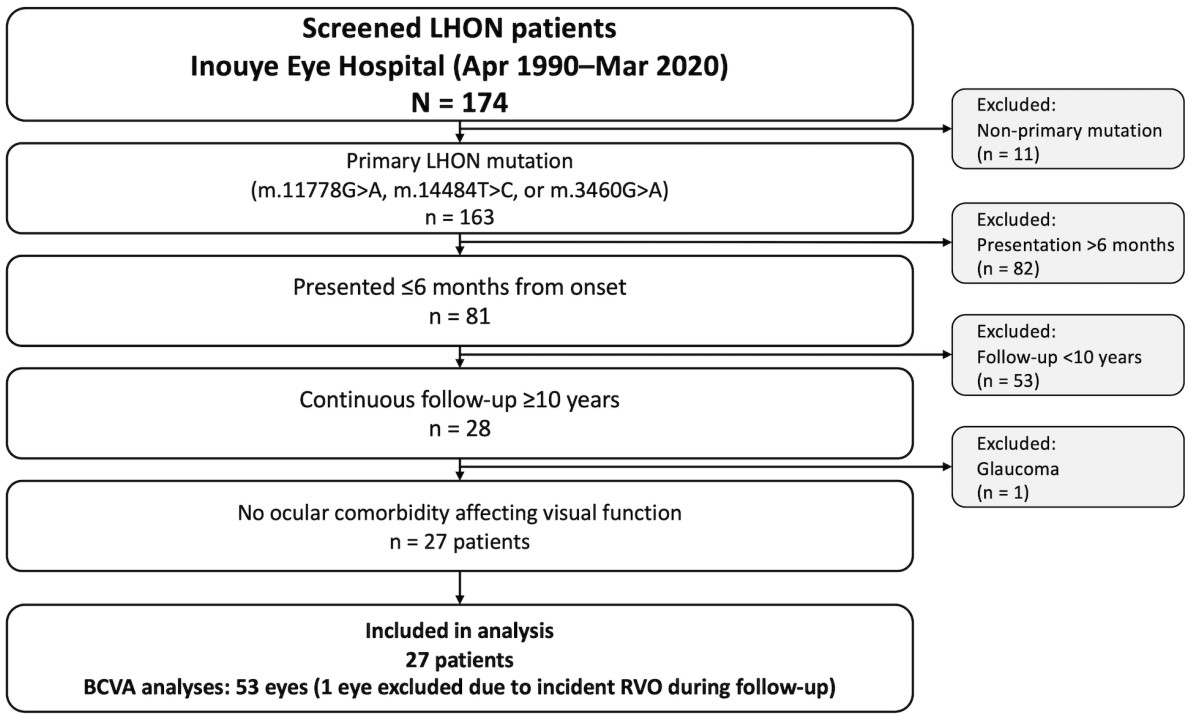

**Fig 1. Patient selection for the 10-year longitudinal LHON cohort.** Of 174 genetically confirmed patients screened, 27 met the inclusion criteria (primary LHON mutation, presentation ≤6 months, and ≥10 years of follow-up). At the eye level, one eye that developed retinal vein occlusion during follow-up was excluded from BCVA analyses (final: 27 patients, 53 eyes).

(e.g., vitamins/supplements including coenzyme Q10 and occasional vasodilators) were used at clinicians' discretion; regimens were heterogeneous and non-protocolized. Thus, our analytic cohort reflects patients with genetically confirmed LHON who presented within 6 months of onset and remained in follow-up at our tertiary center for at least 10 years. It is therefore a long-term follow-up cohort rather than a complete inception cohort of all incident cases.

This study was conducted in accordance with the Declaration of Helsinki and was approved by the Institutional Review Board of Inouye Eye Hospital (approval no. 202508−3). Because this was a retrospective review of existing medical records, the ethics committee waived the requirement for written informed consent. Instead, an opt-out approach was implemented via the hospital website, informing patients about the study and allowing them (or their guardians) to decline participation at any time. Clinical data for this retrospective chart review were accessed for research purposes between September 1, 2025, and October 10, 2025. Although investigators had access to identifiable patient information during data collection, the dataset used for statistical analysis was de-identified and contained no personal identifiers. As the cohort included both adults and minors, the waiver of informed consent and the opt-out procedure extended to parental or guardian permission, as approved by the ethics committee.

Best-corrected visual acuity (BCVA) was measured using a Japanese Snellen chart and expressed in logMAR units. For very low vision, counting fingers, hand motion, and light perception were converted to 2.0, 2.3, and 4.0 logMAR, respectively, consistent with prior LHON natural-history studies and established low-vision quantification methods [5–7]. To assess progression, BCVA was typically assessed at ~4-month intervals within the first 2 years when available, and approximately annually thereafter in routine practice.

To characterize the trajectories of BCVA over 10 years, temporal changes were first visualized descriptively using locally weighted scatterplot smoothing (LOWESS; smoothing parameter [span] = 0.75). This figure is descriptive and not used for formal inference. For inferential analyses, we fitted prespecified piecewise linear mixed-effects models. Based on prior natural-history reports (e.g., REALITY and Newman et al.) [5,6], we defined the Chronic phase as 12–60 months and the Late Chronic phase as 60–120 months. The primary estimand was the slope (logMAR/year) during the Chronic phase, and the secondary estimand was the slope during the Late Chronic phase; that is, we evaluated whether vision tended to improve, worsen, or show no clearly detectable directional trend within each phase. Models included random intercepts for patient (and eye, when applicable) and were adjusted for age at onset, sex, and mtDNA mutation as covariates. In addition, referring to previous Western reports [5,6], we analyzed BCVA progression in a predefined subgroup of patients aged ≥15 years carrying the m.11778G>A mutation. As a sensitivity analysis, we repeated the primary BCVA trajectory models after restricting the cohort to eyes with the m.11778G>A mutation.

We then examined time to attainment of clinically meaningful BCVA thresholds. We selected 1.6 and 1.3 logMAR as thresholds for severe visual impairment [6] and calculated the cumulative proportions of eyes achieving BCVA ≤1.6 and ≤1.3 logMAR during the observation period. Among eyes whose BCVA did not meet these criteria at baseline, time-to-attainment was analyzed using the Kaplan–Meier method. Eyes already meeting the threshold at baseline were excluded from the risk set. Because we did not implement delayed entry (left truncation) in these Kaplan–Meier analyses, early cumulative attainment may be biased if onset-to-enrollment times differ systematically. In addition, curves were estimated per eye without cluster-robust adjustment for within-patient correlation; these analyses are descriptive and intended to complement the linear mixed-effects model results rather than provide independent confirmatory evidence.

Visual fields were assessed using Goldmann perimetry (GP). Patients were included if they underwent GP testing at least once within the first year of onset, at least once in the second year, and at least biennially thereafter. Isopters I-2e, I-4e, and V-4e were primarily used. The severity of visual field defects was graded as follows (S1 Fig): Grade 0, normal field; Grade 1, central scotoma within 10°; Grade 2, central scotoma between 10° and 30°; Grade 3, central scotoma of ≥30°; and Grade 4, only the peripheral visual field remaining. The grade was refined by adding 1.0 point if the scotoma extended more than the prespecified boundary and 0.5 point if it extended halfway or less. Visual fields were independently and blindly reviewed by two neuro-ophthalmologists (Y.T. and A.Y.); disagreements were resolved by consensus. The progression of visual field defects was graded approximately every 6 months during the first 2 years after onset and biennially thereafter.

The critical flicker fusion frequency (CFF) was measured using a Handy Flicker HF-II (NEITZ Co., Ltd., Tokyo, Japan). Following a previously described method [8], measurement was performed in a dark room with fixation on a red target at a 25-cm viewing distance. The threshold was determined by combining ascending and descending methods; after confirming reproducibility, we used the mean of repeated measurements. Patients were included if they underwent CFF measurements at least once in the first year of onset, at least once in the second year, and at least biennially thereafter. To evaluate progression, we assessed CFF approximately every 4 months for the first 2 years after onset and biennially thereafter. CFF is a psychophysical index of visual temporal resolution and has been used as a practical functional marker of afferent visual pathway integrity in optic nerve disease [9]. Prior LHON studies have incorporated CFF improvement into recovery criteria, supporting its role as a complementary endpoint to BCVA and perimetry [10].

As ancillary and exploratory analyses, temporal changes in GP grades and CFF were described using summary statistics and smoothed curves, restricted to subcohorts with repeated measurements over time. Given the small sizes and uneven missingness in these subcohorts, these analyses are positioned as hypothesis-generating and are not used to support the primary conclusions. To reduce information bias, BCVA, GP, and CFF were measured using standardized protocols by experienced orthoptists and ophthalmologists at our institution.

Statistical analyses were performed using R statistical software version 4.4.2 (R Foundation for Statistical Computing, Vienna, Austria). Data are reported as mean±standard deviation (SD) and median (interquartile range). Statistical

significance was set at P < 0.05. We performed available-case analyses without imputation. Linear mixed-effects models inherently accommodate unbalanced repeated measures and used all available observations under a missing-at-random assumption without imputation, and for the Kaplan–Meier analyses, participants lost to follow-up were censored at their last visit.

## Results

Table 1 shows the patients' clinical characteristics in the overall cohort (27 patients; 53 eyes) and subgroup (20 patients; 39 eyes). The mean (SD) age at onset was 32.3 (14.1) and 35.0 (12.5) years in the overall cohort and subgroup, respectively. There were 24 (88.9%) and 17 (85%) males in the overall cohort and subgroup, respectively. In the overall cohort, the m.11778G > A mutation was identified in 24/27 patients (88.9%), m.14484T > C mutation in 3/27 (11.1%), and m.3460G > A mutation in none. Overall, 21/27 patients (77.8%) received at least one non-specific supportive medication during follow-up, vitamins/supplements (18/27, 66.7%) and coenzyme Q10 preparations (16/27, 59.3%); vasodilators were used in 5/27 patients (18.5%) (S1 Table).

Over 10 years, 719 visual acuity assessments were performed. The initial BCVA was 1.45 (0.52) and 1.54 (0.45) log-MAR in the overall cohort and subgroup, respectively. The mean of each patient's worst-recorded BCVA was 1.83 (0.44) and 1.93 (0.26) logMAR in the overall cohort and subgroup, respectively. At 48 months, 26 eyes (49.1%) in the overall cohort and 17 eyes (43.6%) in the subgroup had BCVA ≤1.6 logMAR; 17 eyes (32.1%) in the overall cohort and 8 eyes (20.5%) in the subgroup had BCVA ≤1.3 logMAR. At 120 months, 24 (45.3%) and 15 (38.4%) eyes in the overall cohort and subgroup, respectively, had BCVA ≤1.6 logMAR, while 14 (26.4%) and 6 (15.3%) eyes, respectively, had BCVA ≤1.3 logMAR (Table 2). Detailed descriptive statistics are provided in the supplement (S2 Table).

In the linear mixed-effects model, during the Chronic phase (12–60 months) the overall cohort showed a small but statistically significant improvement of −0.030 logMAR/year (95% CI, −0.045 to −0.015; $t = -3.92$), with a similar direction of effect in the subgroup (−0.025 logMAR/year; 95% CI, −0.044 to −0.007; $t = -2.75$) (Table 3). During the Late Chronic phase (60–120 months), the estimated slopes were close to zero—+0.003 logMAR/year (95% CI, −0.007 to +0.013; $t = 0.59$) in the overall cohort and +0.004 (95% CI, −0.007 to +0.015; $t = 0.72$) in the subgroup—although the confidence intervals were wide and remained compatible with slight long-term improvement or deterioration (Table 3). The LOWESS-smoothed curves (Fig 2) showed gradual recovery after the initial decline and an approximately horizontal course thereafter, consistent with the absence of a clearly detectable late-phase trend at the group level. To address the potential influence of the small m.14484T > C subgroup, we repeated the primary BCVA trajectory analyses restricted to m.11778G > A cases; the phase-specific patterns were similar (S3 Table).

**Table 1. Clinical characteristics and BCVA summary.**

|  |  | All | Subgroup |
|---|---|---|---|
| n |  | 27 | 20 |
| Number of eyes |  | 53 | 39 |
| Age at onset, years | Mean | 32.3 (14.1) | 35.0 (12.5) |
|  | Median | 33.0 (21.0, 43.5) | 33.0 (27.0, 48.2) |
| Male, n (%) |  | 24 (88.9%) | 17 (85%) |
| m.11778 G > A, n (%) |  | 24 (88.9%) | – |
| m.14484 T > C, n (%) |  | 3 (11.1%) | – |

Data are presented as n (%), mean (standard deviation), or median (interquartile range). The subgroup consists of patients with the m.11778G > A mutation who were ≥15 years of age at onset. Abbreviations: BCVA, best-corrected visual acuity.

**Table 2. Representative values of BCVA.**

| | | All | Subgroup |
|---|---|---|---|
| **Initial BCVA,logMAR** | Average | 1.45 (0.52) | 1.54 (0.45) |
| | Median | 1.52 (1.05, 2.00) | 1.52 (1.10, 2.00) |
| **Nadir BCVA,logMAR** | Average | 1.83 (0.44) | 1.93 (0.26) |
| | Median | 2.00 (1.70, 2.00) | 2.00 (1.70, 2.00) |
| **48M BCVA** | ≤ 1.6 logMAR | 26 (49.1%) | 17 (43.6%) |
| | ≤ 1.3 logMAR | 17 (32.1%) | 8 (20.5%) |
| **120M BCVA** | ≤ 1.6 logMAR | 24 (45.3%) | 15 (38.4%) |
| | ≤ 1.3 logMAR | 14 (26.4%) | 6 (15.3%) |

Data are presented as the mean (standard deviation), median (interquartile range), or n (%). The initial BCVA was defined as the visual acuity recorded at the first visit. The nadir BCVA was the worst recorded visual acuity for each eye. The subgroup consists of patients with the m.11778G>A mutation who were ≥15 years of age at onset. Abbreviations: BCVA, best-corrected visual acuity; M, months.

**Table 3. Piecewise linear mixed-effects models (LMM) for BCVA: slopes by predefined phase.**

| Phase (months from onset) | Group | Slope (logMAR/year) | 95% CI | t value |
|---|---|---|---|---|
| **Chronic (12–60)** | Overall | −0.030 | −0.045 to−0.015 | −3.92 |
| | Subgroup | −0.025 | −0.044 to−0.007 | −2.75 |
| **Late Chronic (60–120)** | Overall | 0.003 | −0.007 to +0.013 | 0.59 |
| | Subgroup | 0.004 | −0.007 to +0.015 | 0.72 |

Model: Piecewise LMM with predefined phases (Chronic 12–60 m; Late Chronic 60–120 m). Fixed effects included time within each phase and covariates (age at onset, sex, mtDNA mutation); random intercepts for patient (and eye, if applicable). Negative slopes indicate improvement (lower logMAR), positive slopes indicate worsening. Primary inference emphasizes effect sizes and 95% CIs. Subgroup. "Subgroup" denotes patients aged ≥15 years carrying m.11778G>A.

We estimated the cumulative probability of eyes achieving the prespecified BCVA thresholds over 10 years using Kaplan–Meier time-to-event analysis (Fig 3), defining the event as the first attainment of the threshold. The curve represents the "proportion of non-achievement" (survival function); the cumulative achievement rate is shown as cumulative achievement = 1 − survival. Eyes that had already reached the threshold at baseline were excluded from the risk set. In the overall cohort (A; 42 eyes), the cumulative achievement for BCVA ≤1.6 logMAR was 9.5% (95% confidence interval [CI]: 0.2–18.0), 19.0% (95% CI: 6.3–30.1), 21.4% (95% CI: 8.0–32.9), 26.2% (95% CI: 11.6–38.4), and 31.0% (95% CI: 15.5–43.6) at 12, 24, 36, 48, and 60 months, respectively. Beyond 60 months, the cumulative attainment curves showed little further change, and medians were not reached (survival never fell below 0.5), indicating that new attainments were uncommon during this period. However, interpretation of the late-phase curves is limited by the small number of eyes at risk in the later years. In the subgroup (B; 35 eyes), the proportion was 11.4% (95% CI: 0.2–21.4), 17.1% (95% CI: 3.7–28.7), 20.0% (95% CI: 5.6–32.2), 25.7% (95% CI: 9.7–38.9), and 31.4% (95% CI: 14.2–45.2) at 12, 24, 36, 48, and 60 months, respectively. This rate also remained constant for up to 120 months.

In the overall cohort (C; 49 eyes), the cumulative achievement of BCVA ≤1.3 logMAR was 8.2% (95% CI, 0.2–15.5), 16.3% (5.3–26.1), and 18.4% (6.8–28.5) at 12, 24, and 36 months, respectively, after which it plateaued at 20.4% (8.3–30.9) from 48 through 120 months. In the subgroup (D; 39 eyes), the corresponding probabilities were 7.7% (95% CI, 0.0–15.7) at 12 months and 15.4% (3.3–26.0) at 24 months and then plateaued at 15.4% (3.3–26.0) from 36 through 120 months. Across all curves, the survival function did not fall below 0.5; therefore, the median time to attainment (50%) could not be estimated.

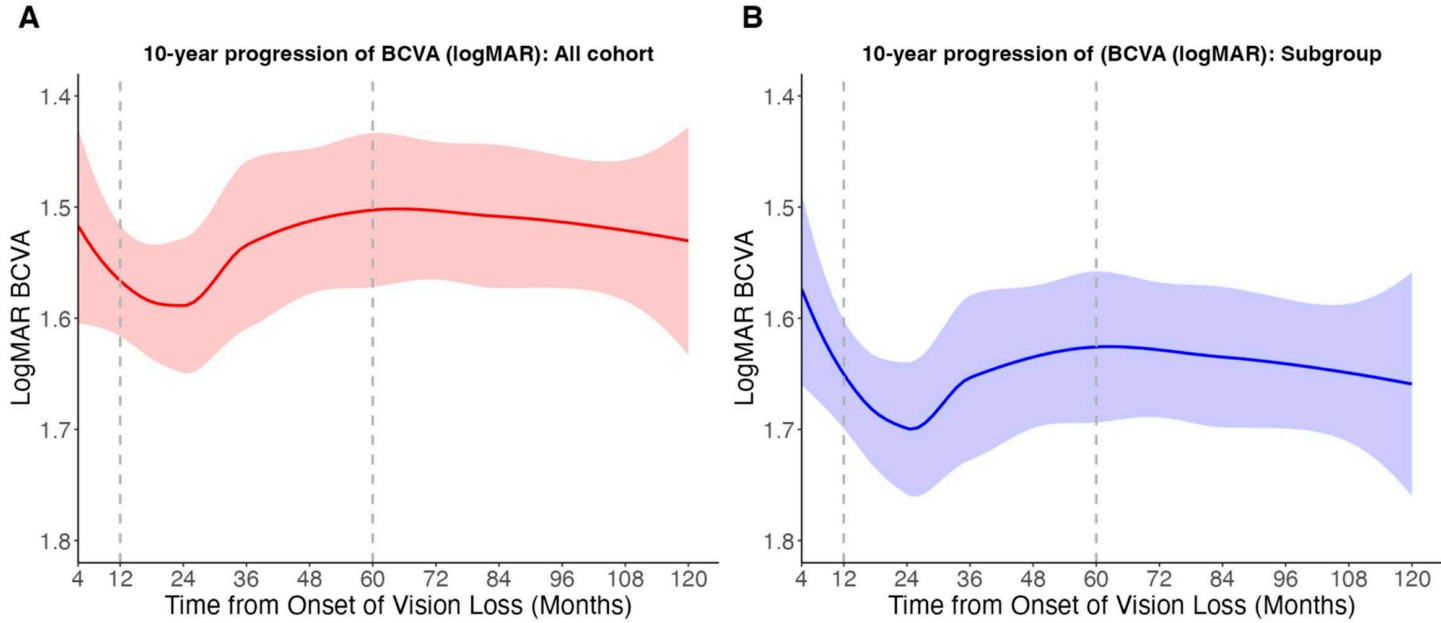

**Fig 2. Ten-year trajectories of BCVA (logMAR) visualized with LOWESS (descriptive only).** A, overall cohort; B, predefined subgroup (≥15 years with the m.11778G>A mutation). Solid lines show LOWESS curves and shaded areas indicate 95% CIs. Vertical dashed lines mark 12 and 60 months (boundaries for Chronic and Late Chronic phases). This figure is descriptive; confirmatory inference is based on piecewise LMMs (Table 3). Lower logMAR indicates better vision.

Regarding visual field defects, 10 patients (19 eyes) were included, and 132 visual field assessments were performed over 10 years. For this group, the mean (SD) age at onset was 38.2 (14.2) years; 17/19 eyes were from males, and the m.11778G>A mutation was present in 15/19 eyes. The GP score averaged 1.11 (0.52) at 6 months and then worsened rapidly, peaking at 2.58 (0.90) at 18 months. Thereafter, the scores stabilized, measuring 2.45 (1.00) at 120 months. Overall, the GP score worsened over the first 1–2 years and then fluctuated within a similar range thereafter. These patterns are compatible with defects becoming established early and tending to persist, but the small sample size and variability preclude firm conclusions about long-term progression or improvement (S2 Fig).

For CFF, 11 patients (19 eyes) were analyzed, and 137 assessments were performed over 10 years. The mean (SD) age of onset for this group was 35.0 (14.6) years; 17/19 eyes were from males, and the m.11778G>A mutation was present in 15/19 eyes. The mean CFF was 20.11 (10.40) Hz at 6 months. By 72 months, CFF gradually recovered to 20.37 (9.87) Hz. The values then plateaued; at 120 months, the mean CFF was 20.11 (10.40) Hz. The mean CFF decreased in the acute stage and then showed some recovery by around 72 months, after which it fluctuated around similar values. This suggests a possible pattern of partial recovery followed by relative stability. However, the small exploratory subgroup and wide variability mean that these findings should be interpreted as descriptive and hypothesis-generating rather than confirmatory (S2 Fig). In addition, a small subset had complete BCVA–GP–CFF data at the prespecified timepoints (6 patients; 11 eyes). Within-eye longitudinal changes across modalities were not always concordant; S4 Table is provided descriptively as hypothesis-generating and was not used to support the primary conclusions.

## Discussion

Here, we characterized long-term visual trajectories in a Japanese LHON cohort with ≥10 years of follow-up at a single tertiary center. BCVA showed a modest improvement during the Chronic phase, whereas late-phase estimates did not

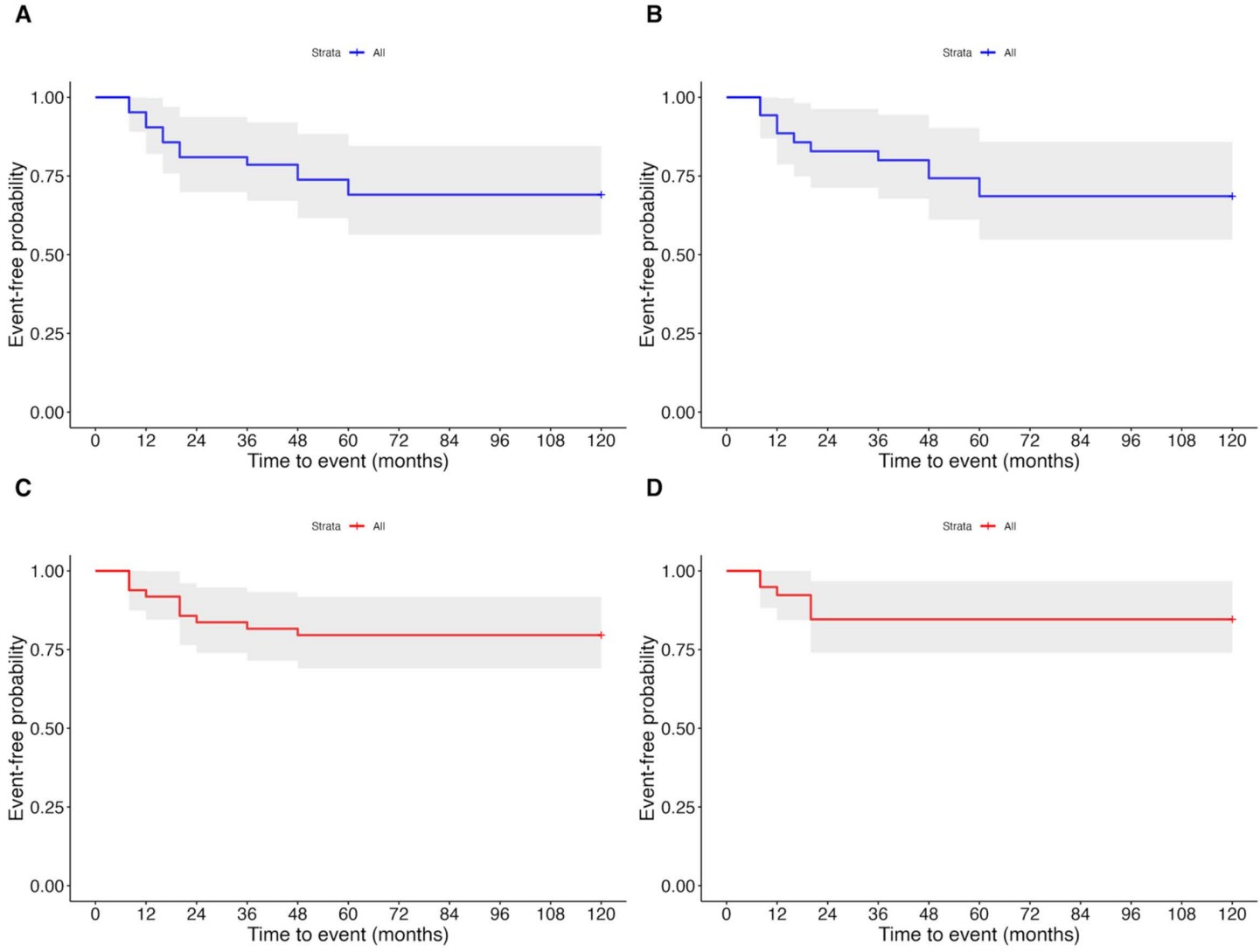

**Fig 3. Time to attainment of visual acuity thresholds: Kaplan–Meier survival curves (proportion of non-achievement).** The curves show the proportion of the survival function: cumulative achievement = 1 − survival. The x-axis represents the number of months since the onset (0 = time of subjective awareness of vision loss). The eyes that met the threshold at baseline were excluded from the risk set. The shaded bands denote 95% confidence intervals. The groups included the overall cohort and subgroups (see Methods for definitions). **(A)** Time to best-corrected visual acuity (BCVA) ≤1.6 logMAR, overall cohort (n = 42 eyes). **(B)** Time to BCVA ≤1.6 logMAR, subgroup (n = 35 eyes). **(C)** Time to BCVA ≤1.3 logMAR, overall cohort (n = 49 eyes). **(D)** Time to BCVA ≤1.3 logMAR, subgroup (n = 39 eyes). Note: In all curves, the survival function did not drop below 0.5; therefore, the median time to attainment (50%) could not be calculated (not reached). Curves depict time to first attainment; eyes at threshold at baseline were excluded.

show a clearly detectable directional trend and should not be interpreted as definitive stability given the wide confidence intervals and the declining number at risk over time. Taken together, these results provide phase-specific quantitative benchmarks for long-term attendees. Although chronic-phase improvement was statistically detectable, the magnitude was small and may have limited functional significance; clinically, recovery—when it occurs—appears typically modest and most likely within the first few years after onset.

The chronic-phase course in our cohort appears different from that reported in large Western cohorts. In the m.11778G>A subgroup of the REALITY study, BCVA declined gradually beyond 18 months of onset [5], and an integrative

analysis likewise suggested a slow decline after the first year [6]. By contrast, we did not detect a clear tendency toward further deterioration between 12 and 120 months in our Japanese cohort, including the predefined subgroup of patients aged ≥15 years with the m.11778G>A mutation. These observed trajectories appear more compatible with limited chronic-phase recovery followed by a relatively flat group-level course. However, direct comparisons are limited by substantial differences in study design, sample selection (in particular, our requirement for ≥10 years of continuous follow-up), follow-up intensity, and treatment exposure, as well as by our modest sample size and wide confidence intervals. The sensitivity analysis restricted to m.11778G>A cases showed similar phase-specific patterns (S3 Table), supporting consistency of the observed trajectories across the mutation types represented in our cohort.

Regarding ethnic differences, favorable visual outcomes have been observed in Asian populations, and covariate-adjusted indirect comparisons have identified Asian ethnicity as an independent factor associated with a better BCVA [11]. Thai and Japanese cohorts have exhibited better outcomes than Western cohorts [12,13]. In our Japanese cohort, the absence of chronic-phase deterioration, including in the subgroup of patients aged ≥15 years with the m.11778G>A mutation, is at least consistent with a pattern reported in some Asian cohorts. However, we cannot fully exclude potential confounding factors, including measurement differences, environmental exposures (such as smoking), and variations in maternal haplogroup distribution. In this context, we regard the present trajectories primarily as single-center Japanese benchmarks that complement large international natural-history datasets and may be useful in future pooled or comparative analyses, for example using meta-regression across diverse cohorts to evaluate population-level differences.

We also explored long-term trajectories of visual field defects and CFF in small ancillary subcohorts. GP scores appeared to worsen over the first 1–2 years and then fluctuated within a similar range thereafter, consistent with the notion that central scotomata become established early and tend to persist. CFF declined early and then showed partial recovery followed by relative stability. However, these analyses were based on only 10 patients for GP and 11 patients for CFF with unbalanced follow-up and substantial variability, and the models were not powered to detect small changes over time. Accordingly, we consider these findings descriptive and hypothesis-generating; they should not be taken as confirmatory evidence of parallel trajectories across visual function modalities. Because repeated BCVA, GP, and CFF measurements were rarely obtained contemporaneously, we analyzed each modality separately and did not model joint patterns of concordant or discordant change.

We did not have protocolized cross-sectional outcome data for patients who did not meet the 10-year follow-up criterion; thus, selection related to long-term follow-up cannot be excluded and the present trajectories should be interpreted as descriptive benchmarks for long-term attendees. Visual improvement has been reported in a subset of LHON patients and early improvers may be underrepresented in long-term follow-up cohorts [14]. Patients with early visual improvement may be underrepresented due to the ≥ 10-year follow-up requirement; thus, our estimates should be interpreted as benchmarks among long-term attendees rather than population-level improvement rates.

This study has several limitations. It was a single-center retrospective cohort from a tertiary eye hospital, which may introduce referral and selection bias and limit generalizability. Follow-up intensity varied and some data were missing; linear mixed-effects models used all available observations under a missing-at-random assumption, but bias remains possible if missingness depended on unobserved outcomes. The sample size was modest, particularly in the late chronic phase and the ancillary GP and CFF subsets, resulting in wide confidence intervals and limited power; therefore, the absence of a statistically significant late-phase slope should not be interpreted as definitive stability, and visit-to-visit fluctuations also related to exertion, or fatigue may further increase measurement variability. Our cohort consisted exclusively of Japanese patients, and genetic or environmental background may differ from Western cohorts; age at onset was somewhat higher than in some earlier studies, consistent with contemporary Japanese epidemiology [15], although long-term follow-up eligibility may also have influenced our age distribution. Finally, because supportive medications were non-protocolized, we did not estimate treatment effects and present findings as descriptive benchmarks rather than

treatment-response estimates. Despite these limitations, the long observation window and genetically confirmed diagnoses provide quantitative benchmarks that may inform future multicenter prospective studies and clinical trials.

Based on a 10-year follow-up of a Japanese LHON cohort with sustained attendance at a single tertiary center, we quantified phase-specific trajectories of BCVA using prespecified piecewise linear mixed-effects models. We observed limited improvement during the Chronic phase (12–60 months) and did not detect a clear directional trend during the Late Chronic phase (60–120 months), although the confidence intervals were wide and compatible with small long-term changes. While the final vision levels were broadly comparable to those reported internationally, our data suggest that, among patients who remain in long-term follow-up, overt late-phase deterioration may not be universal. Rather than establishing ethnic differences, these findings offer descriptive, Japanese single-center benchmarks that may be useful for long-term prognostic counselling and for designing future multicenter studies and clinical trials.

## Supporting information

**S1 Dataset. BCVA long.**
(CSV)

**S2 Dataset. GP long.**
(CSV)

**S3 Dataset. CFF long.**
(CSV)

**S1 Table. Exposure to non-specific supportive medications during follow-up.** Counts are shown as n (%). Supportive medication indicates receipt of at least one of the following: vitamins/supplements (including coenzyme Q10 preparations) and/or vasodilators. Regimens were heterogeneous and not protocolized. No patients received idebenone or gene therapy.
(DOCX)

**S2 Table. Summary of BCVA.** Data are presented as the mean (standard deviation), median (interquartile range). The subgroup consists of patients with the m.11778G>A mutation who were ≥15 years of age at onset. Abbreviations: BCVA, best-corrected visual acuity; M, months.
(DOCX)

**S3 Table. Sensitivity analysis of BCVA slopes restricted to m.11778G>A cases.** Piecewise linear mixed-effects models were refit after restricting the cohort to m.11778G>A cases. Monthly slope estimates were converted to annual rates by multiplying by 12.
(DOCX)

**S4 Table. Descriptive cross-modality trajectories in eyes with complete BCVA, Goldmann perimetry grade, and critical flicker frequency data (6 patients; 11 eyes).** BCVA (logMAR), GP grade, and CFF are shown at 12, 60, and 120 months from onset. Unit of analysis is the eye; some patients contributed both eyes. Because GP and CFF were not protocolized and were typically assessed at ~24-month intervals in routine practice, this table is descriptive and hypothesis-generating, and was not used to support the primary conclusions.
(DOCX)

**S1 Fig. Definition of visual field defect grades in Goldmann perimetry.** Visual field defect grades were defined as follows: Grade 0, normal field; Grade 1, central scotoma within 10°; Grade 2, central scotoma 10–30; Grade 3, central scotoma ≥30°; and Grade 4, only the peripheral field remaining. The grade was further refined by adding 1.0 point if the

scotoma extended more than halfway beyond a prespecified boundary and 0.5 point if it extended halfway or less. Examples are shown for A, Grade 1.0; B, Grade 1.5; C, Grade 2.0; D, Grade 2.5; E, Grade 3.0; and F, Grade 4.0.
(TIF)

**S2 Fig. Ten-year trajectories of visual fields (Goldmann perimetry [GP] score) and critical flicker fusion frequency (CFF): means and locally estimated scatterplot smoothing (LOWESS)-smoothed curves.** Common to all panels. The x-axis shows the months after onset. The y-axis shows the GP score for (A; higher values indicate more severe impairment) and CFF (Hz) for (B). Points and solid lines depict the mean values at each time point, and shaded bands indicate 95% confidence intervals (CIs). LOWESS (local regression) was used for descriptive visualization. See Methods for smoothing parameters (e.g., span). (A) Visual field: LOWESS-smoothed curve with a 95% confidence band (10 patients; 19 eyes; 132 assessments). (B) CFF: LOWESS-smoothed curve with a 95% confidence band (11 patients; 19 eyes; 137 assessments). Note. No inferential modeling was performed for these outcomes.
(TIF)

**S1 Checklist. STROBE checklist for observational cohort studies.**
(DOCX)

## Acknowledgments

We used ChatGPT (GPT-5 Thinking, OpenAI) to assist with language editing and formatting; all authors reviewed and edited the text and take full responsibility for the content.

## Author contributions

**Conceptualization:** Yasuyuki Takai.

**Data curation:** Yasuyuki Takai, Ryoma Yasumoto.

**Formal analysis:** Yasuyuki Takai.

**Investigation:** Yasuyuki Takai, Akiko Yamagami, Mayumi Iwasa, Ryoma Yasumoto.

**Methodology:** Yasuyuki Takai.

**Project administration:** Yasuyuki Takai, Akiko Yamagami, Hitoshi Ishikawa, Masato Wakakura.

**Supervision:** Masato Wakakura.

**Writing – original draft:** Yasuyuki Takai.

**Writing – review & editing:** Akiko Yamagami, Mayumi Iwasa, Kenji Inoue, Hitoshi Ishikawa, Masato Wakakura.

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
