## [Decision Letter · Decision Letter 0]

29 Jan 2026

Dear Dr. Takai,

We look forward to receiving your revised manuscript.

Kind regards,

Jiro Kogo

Academic Editor

PLOS One

Journal Requirements:

2. Please amend the manuscript submission data (via Edit Submission) to include author Yasuyuki Takai.

3. Please include captions for your Supporting Information files at the end of your manuscript, and update any in-text citations to match accordingly. Please see our Supporting Information guidelines for more information: http://journals.plos.org/plosone/s/supporting-information....

4. Please upload a new copy of Figure S1 as the detail is not clear. Please follow the link for more information: https://journals.plos.org/plosone/s/figures

Reviewers' comments:

Reviewer's Responses to Questions

**Comments to the Author**

1. Is the manuscript technically sound, and do the data support the conclusions?

Reviewer #1: Partly

Reviewer #2: Yes

Reviewer #3: Yes

Reviewer #4: Yes

Reviewer #5: Yes

2. Has the statistical analysis been performed appropriately and rigorously?

Reviewer #1: Yes

Reviewer #2: Yes

Reviewer #3: Yes

Reviewer #4: I Don't Know

Reviewer #5: Yes

3. Have the authors made all data underlying the findings in their manuscript fully available?

Reviewer #1: Yes

Reviewer #2: Yes

Reviewer #3: Yes

Reviewer #4: Yes

Reviewer #5: Yes

4. Is the manuscript presented in an intelligible fashion and written in standard English?

Reviewer #1: Yes

Reviewer #2: Yes

Reviewer #3: Yes

Reviewer #4: Yes

Reviewer #5: Yes

Reviewer #1: This study evaluated changes in visual function in patients with LHON who were able to be followed up long-term (over 10 years). Visual function was assessed using BCVA, Goldmann perimetry, and CFF. In conclusion, minimal visual function improvement was observed in the chronic phase, but no improvement occurred in the late chronic phase.

A major issue with retrospective studies like this one is that many patients may not return for follow-up due to visual improvement. Even without long-term follow-up, are there cross-sectional data available? Kim’s study (https://doi.org/10.1080/08820538.2024.2323114), even with a sample size of 11,778, shows a significant number of cases of improvement.

Methods

Why did you analyze 53 eyes instead of 54 eyes out of 27 patients?

Did any of the patients take idebenone as a supplement? Were any of the patients taking other medications that could have influenced the outcome?

Table 1 shows that the average age is somewhat higher than in other studies. Is there a reason for this?

While visual acuity, visual field, and CFF were analyzed separately, a more detailed analysis would be helpful, such as when all three improved, or when one test improved while another worsened.

As the authors describe, the number of cycles for visual field tests and CFF tests is too small to draw any conclusions.

Reviewer #2: I believe that this is an useful addition to the literature concerning the natural history of LHON, and it is worthy of publishing because of the rarity of the disease, the difficulties of assembling a cohort of patients with a long term follow-up and the global necessity of data regarding patients of different ethnicities.

I have only a few comments to make:

lines 115-116 : "Because enrolment occurred after onset, the Kaplan–Meier curves do not incorporate delayed entry"

This is an unclear phrasing. It is true that the K-M curves do not incorporate delayed entries, but that is not caused by the fact that "enrollment occured after onset". Please rephrase.

Also, please include a paragraph about the clinical significance of CFF: what information were the authors supposed to get from this measurement, based on previous data from literature?

line 264 "However, these analyses were based on only 10–11 patients" - this approximation seems odd, you have already stated that you have tested 11 patients

Reviewer #3: General Comments

This study provides rare, long-term (≥10 years) follow-up of Japanese LHON patients, describing phase-specific BCVA trajectories. Strengths include genetically confirmed diagnoses and robust mixed-effects modeling. Limitations include small sample size, selection bias, and limited exploratory analyses. Overall, the manuscript offers valuable descriptive benchmarks for future studies.

Introduction

• Clarity of gap: While the rationale for long-term follow-up is stated, the manuscript could more explicitly emphasize why Asian cohorts might differ from Western cohorts (genetic, environmental, or healthcare-related factors).

• Overlap with previous studies: The introduction references Lam et al. and other natural-history studies but could more clearly differentiate how this study’s design (near-onset enrollment, 10-year follow-up) uniquely contributes new knowledge.

• Objective phrasing: Objectives are clear, but “assess consistency in a predefined ≥15 years, m.11778G>A subgroup” could be rephrased for clarity; it’s slightly ambiguous whether this refers to age, mutation, or both.

Methods

• Selection bias: Only patients with ≥10 years follow-up were included, potentially overrepresenting patients with milder or more stable disease. This could bias results toward apparent late-phase stability.

• Sample size and subgroup limitations: The final analytic cohort (27 patients, 53 eyes) is small, particularly for subgroup and exploratory analyses (GP, CFF), limiting statistical power.

• BCVA conversion: Defining counting fingers, hand motion, and light perception as specific logMAR values is standard, but the rationale could be cited explicitly to justify comparisons over 10 years.

• Kaplan–Meier analyses: Not accounting for delayed entry or clustering by patient may over- or underestimate cumulative probabilities. This should be clearly acknowledged as a limitation in Methods.

• Missing data handling: Linear mixed-effects models assume missing-at-random; it would help to discuss how deviations from this assumption could influence slope estimates, especially given variable follow-up intensity.

Results

• Presentation of slopes: Chronic-phase improvement is statistically significant but small (~0.03 logMAR/year); the clinical relevance should be emphasized.

• Late Chronic phase: Confidence intervals are wide, and the absence of a trend may reflect low power rather than true stability.

• Kaplan–Meier curves: Descriptive, but survival function never falls below 0.5; the lack of median times should be explicitly tied to small sample size and follow-up limitations.

• Exploratory measures (GP, CFF): Data are highly limited; interpretations should remain clearly hypothesis-generating. Presenting mean ± SD without longitudinal modeling may overstate apparent trends.

• Repetition: Some statements about plateauing after 60 months and cumulative probability curves are repeated; streamlining would improve clarity.

Discussion

• Interpretation of ethnic differences: The suggestion that Japanese/Asian patients may have better outcomes is interesting but speculative. Confounding factors (environment, maternal haplogroups, healthcare access) are mentioned but could be emphasized more strongly to avoid overinterpretation.

• Comparison with Western cohorts: Direct comparisons are limited by design differences; this could be highlighted earlier in the discussion.

• Clinical significance: The magnitude of chronic-phase improvement is small; discuss what this means for patient counseling or prognosis.

• Limitations: Well described, but the impact of selection bias on apparent late-phase stability could be reinforced.

• Generalisability: Findings from a single tertiary centre with selected long-term attendees may not apply to broader incident LHON populations. This is noted but could be more strongly linked to the study’s descriptive nature.

Conclusion

BCVA showed modest improvement during the Chronic phase (12–60 months) and no clear trend in the Late Chronic phase (60–120 months). These findings serve as descriptive, phase-specific benchmarks rather than definitive evidence of long-term stability or ethnic differences.

Reviewer #4: This is a nice review of a 10 year f/u cohort of patients with LHON.

It would be helpful to add information about the 3 patients with the 14484 mutation separately, even thought there are only 3 patients with that mutation. Since there may be a large amount of spontaneous improvement in that group it can skew the results and if so, the other 24 patients should be reported separately.

Also, the legend for Figure 1 is too long and merely repeats what is already in the text. This should be shortened.

Reviewer #5: Although LHON treatments (e.g., oral idebenone and gene therapy) have been developed, the rarity of LHON hinders the establishment of appropriate control groups for evaluating efficacy. (Line 60 => I find this remark a bit harsh. The Klopstock and gene therapy studies have control groups.)

Articles relating to these studies should be added.

Wide modifications of the confidence intervals are said to be compatible with small long-term changes. Aren't these simply modifications related to fatigue or the pseudo-Uhthoff syndrome affecting visual acuity that has been reported in LHON?

.

Reviewer #1: No

Reviewer #2: No

Reviewer #3: **Yes:** SARIKA GOPALAKRISHNANSARIKA GOPALAKRISHNANSARIKA GOPALAKRISHNANSARIKA GOPALAKRISHNAN

Reviewer #4: No

Reviewer #5: No

---

## [Author Response · Author response to Decision Letter 1]

13 Feb 2026

Dear Editor,

Thank you for the opportunity to submit a revised version of our manuscript. We have carefully addressed the journal requirements and the reviewers’ comments. Below we summarize how we have complied with the additional journal requirements:

1. PLOS ONE style requirements and file naming

We revised the manuscript to comply with PLOS ONE style requirements, including formatting and file naming. We moved figure captions and table into the main text at first citation (read order) and revised the in-text citation format from superscript numbers to bracketed numbers [ ], consistent with PLOS ONE requirements. We are submitting (i) a clean revised manuscript and (ii) a marked-up version with changes highlighted, along with appropriately named figure and Supporting Information files.

2. Author information (Yasuyuki Takai)

We have amended the manuscript submission data (via “Edit Submission”) to include Yasuyuki Takai as an author, as requested.

3. Supporting Information captions and in-text citations

We included captions for all Supporting Information files at the end of the manuscript and updated the in-text citations to ensure consistency with the Supporting Information labels (S1 Table, S2 Fig, etc.), following the journal’s Supporting Information guidelines.

4. Replacement of S1 Fig

We uploaded a new version of S1 Fig with improved clarity and resolution to meet the figure requirements.

In addition, we updated Figure 1 (study flow) to explicitly report the number of excluded cases by each eligibility criterion, improving transparency of the selection process. We also moved the former Supporting Information Table 1 into the main manuscript as Table 2 to better highlight key results in the Results section and to keep the Supporting Information concise as the number of supplementary items increased.

We also reviewed the reference list to ensure completeness and correctness. We did not intentionally cite any retracted articles.

We appreciate your and the reviewers’ time and consideration, and we hope that the revised manuscript is now suitable for further evaluation.

Sincerely,

Yasuyuki Takai, MD/PhD

on behalf of all authors

Reviewer #1 – Response (green characters)

Comment 1: Selection/attrition bias; cross-sectional data; suggested paper reporting improvement

Response:

Thank you for this important comment. We agree that attrition due to visual improvement is a potential source of selection bias in long-term retrospective cohorts. We have clarified in the Limitations that we did not have protocolized cross-sectional outcome data for patients who did not meet the ≥10-year follow-up criterion and that our results should therefore be interpreted as descriptive benchmarks among long-term attendees. In addition, we added a sentence (with a supporting citation) noting that clinically meaningful visual improvement has been reported in a subset of LHON patients and that early improvers may be underrepresented in long-term follow-up cohorts (Kim et al., 2024).

Manuscript changes:

- Discussion (p20, line 345-352): “We did not have protocolized cross-sectional outcome data for patients who did not meet the 10-year follow-up criterion; thus, selection related to long-term follow-up cannot be excluded and the present trajectories should be interpreted as descriptive benchmarks for long-term attendees. Visual improvement has been reported in a subset of LHON patients and early improvers may be underrepresented in long-term follow-up cohorts [14]. Patients with early visual improvement may be underrepresented due to the ≥10-year follow-up requirement; thus, our estimates should be interpreted as benchmarks among long-term attendees rather than population-level improvement rates.”

Comment 2: Why 53 eyes instead of 54 eyes in 27 patients?

Response:

Among 27 patients (54 eyes), one eye developed retinal vein occlusion during follow-up, which could confound BCVA trajectories. As prespecified in the revision, this eye was excluded from BCVA analyses, resulting in 53 analyzed eyes.

Manuscript changes:

- Materials and methods (p5, lines 83-86) “After applying these patient-level criteria, 27 patients were included. At the eye level, among 54 eyes from these 27 patients, one eye developed retinal vein occlusion during follow-up and was excluded from BCVA analyses, resulting in 53 eyes for the primary analyses (Fig 1).”

Comment 3: Idebenone supplementation and other medications potentially influencing outcomes

Response:

No patients received idebenone or gene therapy during follow-up. Some patients received non-specific/off-label supportive medications (e.g., vitamins/supplements including coenzyme Q10 preparations and/or vasodilators); regimens were heterogeneous and not protocolized. We therefore did not attempt causal inference regarding these medications and clarified this point in the manuscript. Supporting medication exposure is summarized in the Supporting Information.

Manuscript changes:

- Materials and methods (p5, lines 86-89) “No patients received idebenone or gene therapy during follow-up. Low-dose, non-specific off-label supportive medications (e.g., vitamins/supplements including coenzyme Q10 and occasional vasodilators) were used at clinicians’ discretion; regimens were heterogeneous and non-protocolized.”

- Added Supporting Information (S1 Table)

- Results (p11, line185-188) “Overall, 21/27 patients (77.8%) received at least one non-specific supportive medication during follow-up, vitamins/supplements (18/27, 66.7%) and coenzyme Q10 preparations (16/27, 59.3%); vasodilators were used in 5/27 patients (18.5%) (S1 Table).”

emphasized in the Limitations that these data are descriptive and not used to infer treatment effects.

- Discussion (p21, line 365-367):” Finally, because supportive medications were non-protocolized, we did not estimate treatment effects and present findings as descriptive benchmarks rather than treatment-response estimates.”

Comment 4: Mean age appears higher than in other studies—why?

Response:

We acknowledge that the mean age in our analytic cohort may appear higher than in some prior reports. This likely reflects the age distribution of LHON in contemporary Japanese cohorts reported in prior epidemiologic work, and our inclusion criterion requiring ≥10 years of continuous follow-up, which may preferentially capture patients with different healthcare utilization patterns (including older onset and/or long-term attendees at a tertiary center). We have emphasized this as part of the study’s generalizability limitations.

Manuscript changes:

- Discussion (p21, line 361-364): “Our cohort consisted exclusively of Japanese patients, and genetic or environmental background may differ from Western cohorts; age at onset was somewhat higher than in some earlier studies, consistent with contemporary Japanese epidemiology [15], although long-term follow-up eligibility may also have influenced our age distribution.”

Comment 5: Need more detailed “integrated” analysis across BCVA, visual field, and CFF (concordance/discordance)

Response:

We agree that cross-modality concordance and discordance would be informative. However, simultaneous/paired measurements of BCVA, Goldmann perimetry, and CFF at comparable time points were available in only a small subset. Because of this limited overlap, we did not model integrated outcomes formally. Instead, we added a hypothesis-generating descriptive summary limited to eyes with complete data across modalities, clearly labeled as exploratory.

Manuscript changes:

- Result (p21, line 297-299): “Within-eye longitudinal changes across modalities were not always concordant; S4 Table is provided descriptively as hypothesis-generating and was not used to support the primary conclusions.”

- Added a Supporting Information table (S4 table)

Comment 6: Too few cycles for visual field and CFF to draw conclusions

Response:

We agree. This limitation was already stated in the original manuscript (Methods section), and we have further clarified it in the revised version by explicitly noting the limited power/variability in the ancillary GP and CFF subsets (Limitations). These results remain descriptive and hypothesis-generating and were not used to support the primary BCVA trajectory conclusions.

Manuscript changes:

- Discussion (p20, line 357 – p21, line 361): “The sample size was modest, particularly in the late chronic phase and the ancillary GP and CFF subsets, resulting in wide confidence intervals and limited power; therefore, the absence of a statistically significant late-phase slope should not be interpreted as definitive stability, and visit-to-visit fluctuations also related to exertion, or fatigue may further increase measurement variability.”

Reviewer #2 – Response (orange characters)

Comment 1: Unclear phrasing regarding delayed entry in Kaplan–Meier curves (lines 115–116)

Response:

Thank you for pointing this out. We agree that the original phrasing was unclear and could be interpreted as implying that the absence of delayed entry was caused by post-onset enrolment. We have rephrased this sentence to state more precisely that the Kaplan–Meier analyses were conducted without implementing delayed entry (left truncation), and that the resulting curves should be interpreted as descriptive summaries with this limitation in mind.

Manuscript changes:

- Materials and methods (p8, lines 136-138) “Because we did not implement delayed entry (left truncation) in these Kaplan–Meier analyses, early cumulative attainment may be biased if onset-to-enrolment times differ systematically.”

Comment 2: Please include a paragraph about the clinical significance of CFF and its intended information

Response:

We appreciate this suggestion. We have added a dedicated paragraph describing the clinical rationale for CFF in optic neuropathies. Specifically, we clarify that CFF is a psychophysical measure of temporal resolution that reflects afferent visual pathway function, has been used as a functional marker in optic nerve disorders, and has been incorporated as an ancillary outcome in prior LHON literature. We also emphasize that CFF was analyzed as an exploratory/ancillary measure in our study given limited, non-protocolized longitudinal measurements.

Manuscript changes:

- Materials and methods (p9, lines 160-164) “CFF is a psychophysical index of visual temporal resolution and has been used as a practical functional marker of afferent visual pathway integrity in optic nerve disease [9]. Prior LHON studies have incorporated CFF improvement into recovery criteria, supporting its role as a complementary endpoint to BCVA and perimetry [10].”

Comment 3: “Based on only 10–11 patients” is odd; you already state 11 patients tested

Response:

Thank you for catching this inconsistency. We have revised the text to report the sample sizes explicitly and consistently: Goldmann perimetry data were available for 10 patients, whereas CFF data were available for 11 patients. We removed the approximate “10–11” phrasing to avoid ambiguity.

Manuscript changes:

- Discussion (p19, lines 338-339) “However, these analyses were based on only 10 patients for GP and 11 patients for CFF with unbalanced follow-up and substantial variability, and the models…”

Reviewer #3 – Response (purple characters)

General comment

Response:

We sincerely thank the reviewer for the constructive and detailed critique. We agree with the key points regarding the descriptive value of this rare ≥10-year Japanese cohort, as well as the main limitations (small sample size, selection bias, and limited exploratory data for GP/CFF). In the revision, we refined the framing throughout to emphasize that our findings provide phase-specific descriptive benchmarks among long-term attendees, rather than definitive evidence of long-term stability or ethnic differences.

Comment: Clarify why Asian cohorts might differ from Western cohorts

Response:

We agree. We revised the text to more explicitly state plausible reasons why long-term outcomes may differ across cohorts (e.g., genetic background including mitochondrial haplogroups, environmental factors, and healthcare access/utilization), while maintaining a cautious, non-causal tone.

Manuscript changes:

- Introduction (p4, lines 64-66) “Data on long-term visual trajectories in Asian LHON cohorts remain limited, and cross-cohort differences may reflect variation in mitochondrial genetic background (e.g., haplogroup distributions), environmental exposures, and healthcare access.”

Comment: Differentiate this study from previous natural-history studies

Response:

We agree that the unique contribution should be clearer. We revised the Introduction to better distinguish our design—near-onset enrollment with continuous ≥10-year follow-up and phase-specific trajectory modeling using a prespecified piecewise mixed-effects framework—from prior studies with different enrollment windows and/or shorter follow-up.

Manuscript changes:

- Introduction (p4, lines 71- p5 line 75) “In contrast to prior natural-history reports that often include heterogeneous enrolment timing and variable follow-up completeness, our study focuses on a near-onset cohort (presentation within 6 months) with continuous follow-up for ≥10 years at a single tertiary center, enabling phase-specific longitudinal estimates over the chronic and late chronic periods.”

Comment: Objective phrasing is ambiguous (“predefined ≥15 years, m.11778G>A subgroup”)

Response:

Thank you. We clarified that this prespecified subgroup is defined by both onset age ≥15 years and the primary mutation m.11778G>A, to avoid ambiguity.

Manuscript changes:

- Introduction (p4, lines 70-71) “….. and (iii) assess consistency in a predefined subgroup defined as onset age ≥15 years and m.11778G>A subgroup [5].”

Comment: Selection bias (≥10-year follow-up may overrepresent milder/stable disease; may bias toward apparent late-phase stability)

Response:

We agree and have reinforced this limitation. We now emphasize that restricting to ≥10-year continuous follow-up may underrepresent patients who discontinue follow-up (including those with early improvement) and may also enrich for long-term attendees with different disease/healthcare utilization patterns. We therefore frame late-phase “no clear trend” as a descriptive finding in this selected cohort, not proof of true long-term stability.

Manuscript changes:

- Strengthened limitation statements regarding selection/attrition bias and its potential impact on late-phase trajectories.

Comment: Small sample size and subgroup/exploratory limitations (GP/CFF)

Response:

We agree. This is stated in the Materials and methods section in the GP/CFF methods description, where GP/CFF are explicitly characterized as “ancillary and exploratory” and positioned as hypothesis-generating rather than supporting the primary conclusions. The same positioning is carried into the Results, where the GP/CFF analyses are presented descriptively with explicit caution regarding the limited sample size and irregular timing. It is reiterated in the Discussion, which notes the small ancillary subcohorts, limited power, and the descriptive/hypothesis-generating nature of GP/CFF findings.

Manuscript changes:

- No change

Comment: BCVA conversion (CF/HM/LP) should be explicitly justified/cited

Response:

We agree and added an explicit citation supporting the standard logMAR conversions for CF/HM/LP used in longitudinal comparisons. We added the following reference (Reference.7 Lange et al) to support the conversion of the clinical acuity categories “hand motion” and “counting fingers” to logMAR values:

Manuscript changes:

- Materials and methods (p7, lines 112-114). “For very low vision, counting fingers, hand motion, and light perception were converted to 2.0, 2.3, and 4.0 logMAR, respectively, consistent with prior LHON natural-history studies and established low-vision quantification methods [5-7].”

Comment: Kaplan–Meier analyses—lack of delayed entry and clustering should be clearly acknowledged in Methods

Response:

We agree. We clarified that Kaplan

---

## [Decision Letter · Decision Letter 1]

8 Apr 2026

Dear Dr. Takai,

As the corresponding author, your ORCID iD is verified in the submission system and will appear in the published article. PLOS supports the use of ORCID, and we encourage all coauthors to register for an ORCID iD and use it as well. Please encourage your coauthors to verify their ORCID iD within the submission system before final acceptance, as unverified ORCID iDs will not appear in the published article. *Only* the individual author can complete the verification step; PLOS staff the individual author can complete the verification step; PLOS staff the individual author can complete the verification step; PLOS staff the individual author can complete the verification step; PLOS staff *cannot* verify ORCID iDs on behalf of authors.verify ORCID iDs on behalf of authors.verify ORCID iDs on behalf of authors.verify ORCID iDs on behalf of authors.

We look forward to receiving your revised manuscript.

Kind regards,

Jiro Kogo

Academic Editor

PLOS One

Journal Requirements:

Reviewers' comments:

Reviewer's Responses to Questions

**Comments to the Author**

Reviewer #3: All comments have been addressed

Reviewer #4: All comments have been addressed

Reviewer #5: All comments have been addressed

2. Is the manuscript technically sound, and do the data support the conclusions?

Reviewer #3: Partly

Reviewer #4: Yes

Reviewer #5: Yes

3. Has the statistical analysis been performed appropriately and rigorously?

Reviewer #3: Yes

Reviewer #4: I Don't Know

Reviewer #5: Yes

4. Have the authors made all data underlying the findings in their manuscript fully available?

Reviewer #3: Yes

Reviewer #4: Yes

Reviewer #5: Yes

5. Is the manuscript presented in an intelligible fashion and written in standard English?

Reviewer #3: Yes

Reviewer #4: Yes

Reviewer #5: Yes

Reviewer #3: (No Response)

Reviewer #4: (No Response)

Reviewer #5: I thank the authors for all their answers to the various reviewers' questions.

They are perfectly appropriate and make the text much clearer.

.

Reviewer #3: No

Reviewer #4: No

Reviewer #5: **Yes:** Christophe OrssaudChristophe OrssaudChristophe OrssaudChristophe Orssaud

---

## [Author Response · Author response to Decision Letter 2]

9 Apr 2026

Response to Academic Editor and Reviewers

Manuscript ID: PONE-D-25-61898R1

Title: Ten-year natural history of visual function in Japanese patients with Leber hereditary optic neuropathy: a retrospective cohort study

Dear Academic Editor and Reviewers,

Thank you very much for your continued time and consideration of our manuscript. We sincerely appreciate the careful reassessment of our revised manuscript. We are grateful that Reviewer #3, Reviewer #4, and Reviewer #5 indicated in the decision letter that the comments from the previous round had been addressed. We also thank Reviewer #5 for the supportive comment that our responses were appropriate and helped improve the clarity of the manuscript.

In the current round, Reviewer #3 provided additional comments as an attachment. We have carefully considered each point and made specific revisions and clarifications in the manuscript where appropriate. Below, we respond to each comment individually. We also made minor editorial corrections that were identified during this round of revision.

Summary of editorial changes in the current round

In addition to the responses below, the following editorial corrections were made in the current round:

1. Corrected a grammatical error in the Methods section: “we used to prespecify piecewise linear mixed-effects models” was revised to “we fitted prespecified piecewise linear mixed-effects models.”

2. Corrected a minor typographical issue in the Introduction (removed an extraneous space before “phase” in “chronic- phase”).

3. Standardized the American English spelling of “enrollment” throughout the manuscript.

4. Added a brief statement in the Discussion clarifying that the sensitivity analysis restricted to m.11778G>A cases showed similar phase-specific patterns, supporting consistency of the findings across mutation types.

These revisions were editorial in nature and did not affect the study design, statistical analysis, or primary interpretation.

Response to Reviewer #3

We thank Reviewer #3 for the additional comments provided as an attachment for the current revision round. Below, we respond to each point in the order presented.

General

Comment 1: Small sample size and selection bias limit generalizability.

Response: We agree that the modest sample size and potential selection bias are important limitations. In the revised manuscript, these points are explicitly addressed in the following locations:

• Abstract (final sentence): “Given the modest sample size and selection of long-term attendees, these estimates should be interpreted as descriptive, phase-specific benchmarks rather than definitive evidence of long-term stability or ethnic differences.”

• Methods (final paragraph of the cohort description): “Thus, our analytic cohort reflects patients … who presented within 6 months of onset and remained in follow-up at our tertiary center for at least 10 years. It is therefore a long-term follow-up cohort rather than a complete inception cohort of all incident cases.”

• Discussion (Limitations paragraph): We state that the study was a single-center retrospective cohort from a tertiary eye hospital, which may introduce referral and selection bias and limit generalizability, and that the sample size was modest, particularly in the late chronic phase, resulting in wide confidence intervals and limited power.

We believe the manuscript now appropriately frames the findings as descriptive benchmarks rather than population-level estimates.

Comment 2: Wide confidence intervals reduce certainty of long-term trends.

Response: We agree. The wide confidence intervals for the late chronic phase are explicitly acknowledged in the following locations:

• Abstract: “…with wide CIs compatible with small long-term improvement or deterioration.”

• Results (Table 3 and text): The Late Chronic phase slopes are reported with their 95% CIs (+0.003 logMAR/year, 95% CI −0.007 to +0.013 overall), and the text states that “the confidence intervals were wide and remained compatible with slight long-term improvement or deterioration.”

• Discussion: “…late-phase estimates did not show a clearly detectable directional trend and should not be interpreted as definitive stability given the wide confidence intervals and the declining number at risk over time.”

We have used cautious language throughout to avoid overstating the certainty of long-term trends.

Comment 3: Some sentences across sections are overly long, affecting readability.

Response: We have reviewed the manuscript for readability. In the current round, we identified and corrected a grammatical error in the Methods section that contributed to reduced readability (“we used to prespecify” was revised to “we fitted prespecified”). We also corrected a typographical issue in the Introduction (“chronic- phase” → “chronic phase”). We acknowledge that some sentences remain lengthy due to the technical specificity required for statistical methodology; however, we have ensured that each sentence conveys a single main idea and that the overall flow of the manuscript is as clear as possible.

Introduction

Comment: The Introduction could be more concise; it lacks a clear, explicit statement of study hypothesis or primary objective.

Response: In the revised manuscript, the study objectives are now explicitly stated in the final paragraph of the Introduction:

“…we aimed to (i) describe the 10-year natural history of BCVA in a Japanese LHON cohort; (ii) estimate the chronic-phase slope using piecewise LMMs aligned with prior natural-history definitions; and (iii) assess consistency in a predefined subgroup defined as onset age ≥15 years and m.11778G>A subgroup.”

Additionally, we note that this is an observational natural-history study; accordingly, we have stated research aims rather than a formal hypothesis, which we believe is appropriate for this study design. The Introduction was revised in the previous round to improve conciseness and logical flow from background to study objectives.

Methods

Comment 1: Rationale for phase divisions (Acute, Chronic, Late Chronic) should be briefly explained.

Response: The rationale for the phase definitions is provided in the Methods section, where we state:

“Based on prior natural-history reports (e.g., REALITY and Newman et al.) [5,6], we defined the Chronic phase as 12–60 months and the Late Chronic phase as 60–120 months.”

These references (the REALITY study and Newman et al.) provide the natural-history framework from which the phase boundaries were derived. We believe this is sufficient justification for the predefined phase divisions.

Comment 2: Inclusion/exclusion criteria and handling of missing data are not fully detailed.

Response: The inclusion and exclusion criteria are specified in the Methods section. Specifically:

• Inclusion criteria: primary LHON mutation (m.11778G>A, m.14484T>C, or m.3460G>A), presentation within 6 months of visual loss onset, no ocular comorbidity likely to affect visual function, and continuous follow-up for ≥10 years with at least annual BCVA measurements.

• Exclusion: one eye was excluded due to retinal vein occlusion during follow-up (Fig 1).

• Missing data: The Methods section states: “We performed available-case analyses without imputation. Linear mixed-effects models inherently accommodate unbalanced repeated measures and used all available observations under a missing-at-random assumption without imputation, and for the Kaplan–Meier analyses, participants lost to follow-up were censored at their last visit.”

We believe the handling of missing data is now clearly described.

Comment 3: Dense sentences reduce readability; consider splitting.

Response: As noted above under General Comment 3, we have corrected the grammatical error in the Methods section and improved readability where possible. We have also reviewed other dense passages for clarity.

Comment 4: Statistical software/package not specified.

Response: The statistical software is specified in the Methods section:

“Statistical analyses were performed using R statistical software version 4.4.2 (R Foundation for Statistical Computing, Vienna, Austria).”

Results

Comment 1: What is Nadir BCVA? 48M and 120M?

Response: The definition of nadir BCVA is provided in the footnote of Table 2:

“The nadir BCVA was the worst recorded visual acuity for each eye.”

This is an eye-level definition: nadir BCVA represents the single worst BCVA measurement observed for each eye during the entire follow-up period. The values at 48M and 120M are not nadir values; rather, they represent the proportions of eyes achieving the prespecified BCVA thresholds (≤1.6 and ≤1.3 logMAR) at those time points. We have rechecked the manuscript and confirmed that the distinction between nadir BCVA (worst-ever) and time-specific BCVA proportions is clearly presented in Table 2 and the accompanying text.

Comment 2: Dense numeric reporting; tables or figures could improve clarity.

Response: In the revised manuscript, key numeric results are summarized in tables:

• Table 1: Clinical characteristics and BCVA summary

• Table 2: Representative values of BCVA (initial, nadir, 48M, 120M proportions)

• Table 3: Piecewise LMM slopes by predefined phase

Detailed descriptive statistics are provided in S2 Table. The Kaplan–Meier cumulative achievement estimates are presented graphically in Fig 3. We believe the current combination of tables, figures, and text provides a clear presentation of the results.

Comment 3: Percentages of eyes vs. patients sometimes unclear.

Response: We have rechecked the manuscript to ensure that percentages are consistently attributed to the correct unit of analysis. In particular:

• Table 1 presents patient-level characteristics (e.g., 24/27 patients [88.9%] were male).

• Table 2 and the BCVA proportion analyses are presented at the eye level (e.g., 26 eyes [49.1%]).

• The Results text now consistently specifies “eyes” or “patients” when reporting percentages.

We believe the distinction is now clearly presented throughout.

Comment 4: Figures (LOWESS, Kaplan–Meier) need clearer referencing and brief interpretation in text.

Response: The revised manuscript includes in-text references to all figures:

• The LOWESS curves (Fig 2) are described as: “The LOWESS-smoothed curves (Fig 2) showed gradual recovery after the initial decline and an approximately horizontal course thereafter, consistent with the absence of a clearly detectable late-phase trend at the group level.”

• The Kaplan–Meier analyses (Fig 3) include detailed cumulative achievement estimates in the Results text, with explicit in-text references to each panel (A–D).

We believe the figures are now adequately referenced and briefly interpreted in the text.

Comment 5: Functional significance of BCVA changes could be emphasized.

Response: The functional significance of the observed BCVA changes is addressed in the Discussion:

“Although chronic-phase improvement was statistically detectable, the magnitude was small and may have limited functional significance; clinically, recovery—when it occurs—appears typically modest and most likely within the first few years after onset.”

Additionally, the BCVA thresholds used in the Kaplan–Meier analyses (≤1.6 and ≤1.3 logMAR) were selected to represent clinically meaningful levels of severe visual impairment, as referenced from Newman et al. [6]. We believe the functional significance is now appropriately discussed.

Comment 6: Consider emphasizing functional significance (e.g., BCVA ≤1.6 logMAR = threshold for legal blindness in Japan).

Response: We appreciate this suggestion. We note that the thresholds used in our study (≤1.6 and ≤1.3 logMAR) were selected based on prior LHON natural-history literature [6] to represent clinically meaningful levels of severe visual impairment, rather than the legal blindness criteria specific to Japan. While the Japanese legal blindness threshold is relevant, introducing a country-specific legal definition may complicate the discussion beyond the scope of this natural-history study that aims to provide internationally comparable benchmarks. We have retained the current thresholds and their clinical interpretation as defined in the Methods section.

Discussion

Comment 1: There could be improvement in visual acuity after certain time period across all LHON types, which was not addressed adequately.

Response: We agree that this is an important point. In the revised manuscript, the primary piecewise LMM analysis was performed in the overall cohort (which includes both m.11778G>A and m.14484T>C cases) and in the predefined m.11778G>A subgroup. To directly address the concern about consistency across mutation types, we performed a sensitivity analysis restricted to m.11778G>A cases (S3 Table), and the Results text states:

“To address the potential influence of the small m.14484T>C subgroup, we repeated the primary BCVA trajectory analyses restricted to m.11778G>A cases; the phase-specific patterns were similar (S3 Table).”

In the current round, we have added the following sentence to the Discussion to more explicitly address this point:

“The sensitivity analysis restricted to m.11778G>A cases showed similar phase-specific patterns (S3 Table), supporting consistency of the observed trajectories across the mutation types represented in our cohort.” [NEW]

We acknowledge that because our cohort contained no m.3460G>A cases and only 3 patients with m.14484T>C, we cannot provide mutation-specific trajectory estimates for all three primary LHON mutations. This limitation is inherent to the rarity of LHON and the composition of our cohort. We have stated this in the Limitations section.

Comment 2: Some interpretations may overstate findings; use more cautious phrasing.

Response: We have reviewed the manuscript and used cautious language throughout. Key examples include:

• Abstract: “…we did not detect a clear directional trend thereafter” (rather than claiming stability)

• Discussion: “…late-phase estimates did not show a clearly detectable directional trend and should not be interpreted as definitive stability”

• Discussion: “…overt late-phase deterioration may not be universal” (rather than claiming no deterioration occurs)

• Conclusions: “Rather than establishing ethnic differences, these findings offer descriptive, Japanese single-center benchmarks”

We believe the current phrasing is appropriately cautious throughout.

Comment 3: Limited discussion on potential biological/environmental reasons for differences from Western cohorts.

Response: This point is addressed in the Discussion, where we state:

“However, we cannot fully exclude potential confounding factors, including measurement differences, environmental exposures (such as smoking), and variations in maternal haplogroup distribution.”

Additionally, the Introduction notes that:

“…cross-cohort differences may reflect variation in mitochondrial genetic background (e.g., haplogroup distributions), environmental exposures, and healthcare access.”

We acknowledge that a more extensive discussion of potential biological mechanisms would be informative but would be speculative given our observational data. We have framed the observed differences as descriptive rather than mechanistic, stating that our results serve as “single-center Japanese benchmarks that complement large international natural-history datasets.”

Comment 4: Clinical relevance of small chronic-phase improvements could be elaborated.

Response: The clinical relevance is discussed in the Discussion:

“Although chronic-phase improvement was statistically detectable, the magnitude was small and may have limited functional significance; clinically, recovery—when it occurs—appears typically modest and most likely within the first few years after onset.”

Furthermore, the Kaplan–Meier analyses provide complementary information about the clinical meaningfulness of BCVA changes by showing the proportions of eyes achieving clinically meaningful thresholds over time. The plateau in cumulative attainment after 60 months further supports the interpretation that most clinically meaningful recovery occurs in the earlier chronic period. We believe these discussions adequately address the clinical relevance.

Response to Reviewer #4

We sincerely thank R

---

## [Editor Report · Decision Letter 2]

12 Apr 2026

Ten-year natural history of visual function in Japanese patients with Leber hereditary optic neuropathy: a retrospective cohort study

PONE-D-25-61898R2

Dear Dr. Takai

We’re pleased to inform you that your manuscript has been judged scientifically suitable for publication and will be formally accepted for publication once it meets all outstanding technical requirements.

Kind regards,

Jiro Kogo

Academic Editor

PLOS One
---

## [Editor Report · Acceptance letter]

PONE-D-25-61898R2

PLOS One

Dear Dr. Takai,

I'm pleased to inform you that your manuscript has been deemed suitable for publication in PLOS One. Congratulations! Your manuscript is now being handed over to our production team.

Kind regards,

on behalf of

Prof. Jiro Kogo

Academic Editor

PLOS One